# TOWARDS RELIABLE TRANSFERABILITY OF TARGETED ADVERSARIAL ATTACKS AGAINST MODEL DISCREPANCY

## ABSTRACT

Adversarial examples pose serious threats to deep neural networks, particularly in black-box settings where transferability plays a key role. While non-targeted transfer attacks have advanced significantly, achieving reliable *targeted* transfer—forcing predictions to a specific label on an unseen model—remains far more challenging. We trace this difficulty to three major surrogate–target discrepancies: feature-extractor mismatch, classifier sensitivity mismatch, and decision-boundary misalignment. We formalize targeted transfer as a robust optimization problem over these uncertainties and propose a tractable relaxation, *Targeted Attack toward Reliable Transferability (TART)*. TART integrates three components: (i) *Expectation over Transformation* to promote robustness under feature-extractor variability; (ii) *Latent Mixing* to handle classifier sensitivity differences; and (iii) *Feature Alignment* with a representative exemplar to mitigate decision-boundary shifts. Extensive experiments on ImageNet and CIFAR-10 show that TART significantly outperforms state-of-the-art targeted transfer attacks across CNN and Vision Transformer architectures. For example, when transferring from ResNet-50 to Swin-S on ImageNet, TART achieves a 42.7% higher attack success rate than the strongest baseline.

## 1 INTRODUCTION

Adversarial examples (Szegedy et al., 2014; Goodfellow et al., 2015) are inputs perturbed to mislead deep neural networks (DNNs), posing serious risks in safety-critical domains such as face recognition, autonomous driving, and medical diagnosis (Eykholt et al., 2018; Finlayson et al., 2019). Attacks are typically categorized as *white-box*, where model parameters are known, or *black-box*, where they are not. Black-box attacks are especially concerning, as real-world systems rarely expose internal details yet remain vulnerable due to *transferability*—adversarial examples crafted on a surrogate model often succeed on unseen targets (Papernot et al., 2016).

While transferability enables strong black-box *non-targeted* attacks—where predictions change to any incorrect label (Dong et al., 2018; Wang et al., 2024; Chen et al., 2023; Li et al., 2025)—achieving reliable *targeted* transfer, where predictions are forced to a specific label, remains far more challenging. Recent methods (Wei et al., 2023; Byun et al., 2023; Liang et al., 2025) still fall well short of non-targeted performance.

We attribute this difficulty to fundamental discrepancies between the surrogate and target classification models. Each classification model can be regarded as comprising two components: a *feature extractor*, which maps the input into a high-dimensional representation, and a *classifier*, which maps this representation to class logits. Targeted transfer requires guiding the perturbed input so that its features lie in the target model's decision region for the desired class, leaving little tolerance for mismatches. We identify three main sources of discrepancy: (i) *Feature-extractor mismatch:* Architectural and training differences yield distinct feature spaces and invariances, misaligning extracted representations even for the same input. (ii) *Classifier sensitivity mismatch:* Even if the extracted features align, a perturbation that drives the surrogate classifier to predict the target label may fail on the target classifier due to differences in how features are mapped to logits (e.g., Jacobians). (iii) *Decision-boundary*

*mismatch:* Variations in class decision surfaces mean that moving an input into the surrogate's target class region does not guarantee it will lie within the target model's region for that class.

We cast these discrepancies as structured uncertainty in mapping inputs to target logits. Specifically, we consider uncertainty over: (i) feature-extractor variability, (ii) classifier-level Jacobian mismatch, and (iii) local shifts in class decision regions. We then pose an ideal robust objective: maximize the worst-case target-class margin under this uncertainty, ensuring adversarial perturbations remain effective despite model differences. While intractable to solve exactly, this perspective motivates a principled relaxation forming the basis of our method, *Targeted Attack toward Reliable Transferability (TART)*.

TART implements this relaxation via three complementary components:

- *Expectation over Transformation (EoT):* By optimizing under diverse input transformations (e.g., rotations, flips, crops), we approximate variability in feature extractors and promote feature robustness.
- *Latent Mixing:* To address classifier sensitivity mismatch, we interpolate between clean and adversarial features while optimizing for the target label. This makes the perturbation effective even when the target classifier remains partially influenced by benign features, compensating for differences in feature-to-logit mappings.
- *Feature Alignment:* To mitigate decision-boundary variations, we align adversarial features within the surrogate model to those of a representative exemplar from the target class. This guidance steers adversarial examples toward semantically central and architecture-consistent regions of the target class, reducing reliance on the surrogate's exact boundaries.

Together, these components operationalize our robust optimization perspective in a tractable form. We also provide theoretical analysis showing that, under standard local assumptions, our relaxation provides a provable lower bound on the worst-case targeted loss.

Our key contributions are as follows:

- *Structured view of targeted transfer difficulty.* We identify three critical surrogate–target discrepancies—feature-extractor mismatch, classifier sensitivity (Jacobian) mismatch, and decision-boundary misalignment—and frame targeted transfer as a robust optimization problem over a composite uncertainty set.

- *Principled objective with practical implementation.* We introduce a worst-case margin objective under this uncertainty set and design TART, a tractable relaxation realized through three components: Expectation over Transformation, Latent Mixing, and Feature Alignment.

- *Theoretical guarantee and insight.* We prove that TART provides a lower bound on the worst-case targeted loss under standard local assumptions and analyze how its components influence robustness.

- *State-of-the-art empirical results.* Experiments on ImageNet and CIFAR-10 show that TART substantially outperforms prior targeted transfer attacks across CNN and Transformer architectures—e.g., on ImageNet, ResNet-50 $\rightarrow$ Swin-S transfer achieves 65.7% success vs. 23.0% for the strongest baseline.

## 2 RELATED WORK

Transfer-based attacks leverage the cross-model effectiveness of adversarial examples (Papernot et al., 2016). Non-targeted transfer—where any incorrect label suffices—has been widely explored through gradient-based optimization (Dong et al., 2018; Lin et al., 2020; Wang & He, 2021), input transformations (Xie et al., 2019; Dong et al., 2019; Zou et al., 2020; Wang et al., 2024), and model ensembling (Liu et al., 2017; Chen et al., 2023; Li et al., 2025). In contrast, *targeted* transfer, which enforces a specific label, is far more challenging. Existing approaches mainly include: (i) *loss design*, e.g., Po+Trip (Li et al., 2020) adds triplet regularization and Poincaré distance, while logit attacks (Zhao et al., 2021) optimize logits directly; (ii) *input transformations*, such as ODI (Byun et al., 2022) using 3D rendering and SU (Wei et al., 2023) enforcing global–local consistency; (iii) *feature-space alignment*, including AA (Inkawhich et al., 2019) minimizing feature gaps, CFM (Byun et al., 2023) mixing features, and FTM (Liang et al., 2025) adding learnable perturbations; and

(iv) *generation-based attacks* (Yang et al., 2022; Wang et al., 2023; Fang et al., 2024), which learn target-class distributions but require full training data and auxiliary networks, limiting practicality. Despite these advances, targeted transfer lags far behind non-targeted transfer due to the three major surrogate–target discrepancies discussed in Section 1: feature-extractor mismatch, classifier sensitivity differences, and decision-boundary variations.

## 3 MODEL DISCREPANCIES IN TARGETED TRANSFERABILITY

### 3.1 PROBLEM SETUP

Let $x \in \mathbb{R}^d$ be an input and $\|\delta\| \leq \varepsilon$ a norm-bounded perturbation. A $C$-class classification model is modeled as a composition of a *feature extractor* and a *classifier*:

$$f(x) = g\big(h(x)\big), \qquad h : \mathbb{R}^d \to \mathbb{R}^p, \ \ g : \mathbb{R}^p \to \mathbb{R}^C, \tag{1}$$

with surrogate and target models denoted $f_s = g_s \circ h_s$ and $f_t = g_t \circ h_t$. For $u = f(x) \in \mathbb{R}^C$ and target label $\tau \in \{1, \ldots, C\}$, define the *targeted margin*:

$$m_\tau(u) = u_\tau - \max_{k \neq \tau} u_k. \tag{2}$$

A targeted attack aims to find $\delta$ such that $m_\tau\big(f_t(x + \delta)\big) > 0$, i.e., $x + \delta$ is classified as $\tau$ on the unknown target model.

### 3.2 FORMALIZATION OF MODEL DISCREPANCIES

As mentioned in Section 1, the difficulty of reliable targeted transfer stems from structural differences between the surrogate and target models. We propose to characterize these differences through three types of mismatches, detailed below.

**D1: Feature-extractor mismatch.** The first source of discrepancy arises from differences in the feature extractors, which map inputs to a high-dimensional feature space that serves as input to the subsequent classifier. Such mismatch may result not only from architectural or training-scheme variations (e.g., CNN vs. Transformer, data augmentations, normalization strategies) but also from inherent stochasticity in training, which can produce different feature spaces even under the same architecture and scheme. These factors impose distinct invariances on the feature space, causing the extracted representations of the same input to misalign across models. This misalignment poses a key challenge for transferring adversarial examples between models.

To capture this discrepancy, we introduce an uncertainty term modeling the difference between the outputs of the two extractors:

$$h_t(x) = h_s(x) + \Delta_h(x), \quad \|\Delta_h(x)\| \leq \rho_h, \tag{3}$$

where $\Delta_h(x)$ represents the feature discrepancy and $\rho_h$ bounds its magnitude.

**D2: Classifier sensitivity mismatch.** Even if the extracted features from the surrogate and target extractors align well, the classifiers may still behave differently. A classifier maps features to logits, and its sensitivity to small feature changes is characterized by its Jacobian. When the surrogate and target classifiers differ in this mapping, a perturbation that forces the surrogate classifier to predict the target label may fail on the target classifier because their feature-to-logit relationship varies.

We model this discrepancy by introducing an uncertainty term that captures variations in classifier behavior. Specifically, we assume the target classifier's logits relate to those of the surrogate through a perturbed mapping:

$$g_t\big(h_s(x + \delta)\big) = g_s\big(h_s(x + \delta)\big) + \Delta J \big(h_s(x + \delta) - h_s(x)\big) + b + r_g, \tag{4}$$

where $\Delta J$ represents the Jacobian mismatch, $b$ is a logit offset, and $r_g$ is a higher-order remainder term accounting for local curvature. We bound these as $\|\Delta J\| \leq \rho_J$ and $\|b\|_\infty \leq \rho_b$.

**D3: Decision-boundary mismatch.** The third source of discrepancy arises from differences in the decision boundaries between the surrogate and target models. Even when features align and classifier sensitivities are similar, the exact shape and location of class regions can differ across models. As

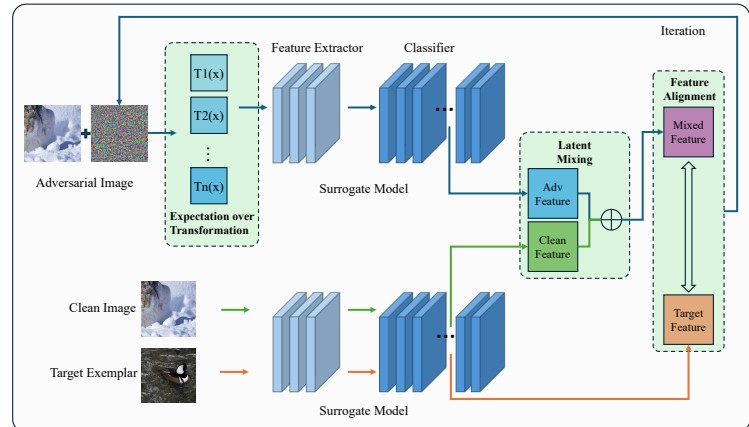

Figure 1: Overview of our TART attack framework.

a result, a perturbation that moves an input into the target class region of the *surrogate* does not guarantee it will also lie inside the target class region of the *target model*. This misalignment is especially problematic for targeted attacks, which require steering the input into a very specific class region, leaving little tolerance for boundary shifts.

We capture this mismatch using a margin bias term $\beta$ that accounts for local shifts between the surrogate and target decision surfaces. Formally, the target margin may be offset from the surrogate's by up to $\beta$:

$$m_\tau(g_t(z)) \geq m_\tau(g_s(z)) - \beta, \quad |\beta| \leq \rho_\beta, \tag{5}$$

where $\beta$ captures this potential margin difference and $\rho_\beta$ bounds its magnitude. Even small shifts can invalidate a targeted perturbation that appears successful on the surrogate.

**Composite uncertainty set.** We define the composite uncertainty set $\mathcal{U}(\rho)$ to jointly capture the deviations between the surrogate and target models discussed above. It is given by:

$$\mathcal{U}(\rho) = \left\{ (\Delta_h, \Delta J, b, \beta) : \|\Delta_h\| \leq \rho_h,\ \|\Delta J\| \leq \rho_J,\ \|b\|_\infty \leq \rho_b,\ |\beta| \leq \rho_\beta \right\}. \tag{6}$$

A transferable adversarial perturbation must therefore be robust to all variations represented in $\mathcal{U}(\rho)$.

### 3.3 IDEAL ROBUST OBJECTIVE

We now formalize an ideal robust objective that explicitly accounts for model discrepancies. The goal is to find a perturbation $\delta$ that maximizes the *worst-case* targeted margin under all allowable variations in the uncertainty set $\mathcal{U}(\rho)$. Formally, we define the robust margin for a candidate $\delta$ as:

$$\text{RobMarg}(x, \delta; \rho) := \inf_{(\Delta_h, \Delta J, b, \beta) \in \mathcal{U}(\rho)} \left[ m_\tau\big(g_s(h_s(x+\delta) + \Delta_h) + \Delta J(h_s(x+\delta) - h_s(x)) + b) - \beta \right], \tag{7}$$

where $m_\tau$ is the targeted margin function from Eq. 2. The inner infimum captures the worst-case effect of feature misalignment ($\Delta_h$), Jacobian mismatch ($\Delta J$), logit shifts ($b$), and local decision-boundary uncertainty ($\beta$).

The ideal robust optimization problem is then:

$$\max_{\|\delta\| \leq \varepsilon} \text{RobMarg}(x, \delta; \rho). \tag{8}$$

This objective formalizes targeted transfer as maximizing a worst-case targeted margin over the composite uncertainty set $\mathcal{U}(\rho)$.

## 4 TARGETED ATTACK TOWARD RELIABLE TRANSFERABILITY (TART)

Directly optimizing the robust objective in Eq. 8 is intractable: the composition $g_s \circ h_s$ is highly nonconvex, and the inner infimum over $(\Delta_h, \Delta J, b, \beta) \in \mathcal{U}(\rho)$ couples multiple sources of model

mismatch. We therefore design *Targeted Attack toward Reliable Transferability (TART)* as a tractable relaxation of Eq. 8. Rather than attempting to solve the max–min problem exactly, TART decouples the three discrepancy types (D1–D3) and replaces their inner worst cases with estimable terms that can be optimized with stochastic gradient methods.

### 4.1 RELAXING THE ROBUST OBJECTIVE

In Eq. 7, the robust margin RobMarg is expressed as an infimum over feature-extractor mismatch $\Delta_h$, classifier/Jacobian mismatch $(\Delta J, b)$, and decision-boundary bias $\beta$, all constrained by the uncertainty set $\mathcal{U}(\rho)$. Our design of TART follows this structure: each component is introduced to approximate the worst-case effect of one type of mismatch in a way that is computationally feasible.

- **Extractor mismatch $\Delta_h$ (D1) $\rightarrow$ Expectation over Transformation (EoT).** We approximate variability in the unknown extractor $h_t$ by optimizing the loss under random input-space transformations that induce variability in the surrogate extractor $h_s$, leading to an Expectation-over-Transformation objective over $t \sim \Pi$.
- **Classifier mismatch $(\Delta J, b)$ (D2) $\rightarrow$ Latent Mixing.** We reduce sensitivity to differences in feature-to-logit mappings by stochastically mixing clean and adversarial activations within the surrogate classifier, effectively shortening the feature displacement that appears in the term $\Delta J(h_s(x + \delta) - h_s(x)) + b$.
- **Decision-boundary bias $\beta$ (D3) $\rightarrow$ Feature Alignment.** We mitigate local shifts in decision boundaries by steering adversarial features toward the direction of a target-class exemplar, promoting progress along directions that locally increase the target margin and are consistent across architectures.

The next three subsections describe these components in detail. We then combine them into the TART objective in Eq. 14. In Section 5, we analyze a margin-based counterpart of this objective and show that, under mild assumptions, it provides a provable lower bound on the robust targeted margin $\text{RobMarg}(x, \delta; \rho)$, thereby formally linking TART to the ideal robust formulation.

### 4.2 EXPECTATION OVER TRANSFORMATION (EoT)

EoT optimizes the perturbation over a set of input transformations, approximating feature-extractor variability. This is achieved by randomly applying different transformations to the input and computing the adversarial loss for each transformed version; the objective is then averaged across all transformations to capture uncertainty in the feature extractor.

Let $\mathcal{T} = \{\text{rotation, flip, translation, scale, crop}\}$ denote a pool of *input-space* transformations that are applied *before* the feature extractor $h_s$. At each attack iteration, we randomly select two transformations $t_1, t_2 \in \mathcal{T}$ and apply their composition to the perturbed input, yielding $t(x + \delta) = t_2 \circ t_1(x + \delta)$. We repeat this process $N$ times per iteration (sampling different pairs), obtaining transformations $\{t^{(i)}\}_{i=1}^{N}$ and the Monte-Carlo estimator

$$\mathcal{J}_{\text{EoT}}(x, \delta) \;=\; \frac{1}{N} \sum_{i=1}^{N} \mathcal{L}\Big(g_s\big(h_s(t^{(i)}(x + \delta))\big), \tau\Big) \;\approx\; \mathbb{E}_{t \sim \Pi}\big[\mathcal{L}\big(g_s(h_s(t(x + \delta))), \tau\big)\big], \quad (9)$$

where $\Pi$ is the uniform distribution over the composed transformations and we use a logit-based loss $\mathcal{L}$ to avoid saturation.

From the perspective of the robust margin in Eq. 7, this EoT term approximates robustness to feature-extractor mismatch $\Delta_h$: sampling $t \sim \Pi$ induces variability in $h_s(t(\cdot))$ that mimics plausible deviations between $h_s$ and $h_t$, addressing discrepancy D1. Section 5 makes this connection precise via a Lipschitz-based lower bound.

### 4.3 LATENT MIXING

Latent Mixing addresses classifier sensitivity mismatch by stochastically blending activations from clean and adversarial inputs across multiple layers within the classifier. This helps ensure that the perturbation remains effective even when the target classifier exhibits different feature-to-logit sensitivities.

Concretely, for each sampled transformation $t$ and each training iteration:

1. Sample a Bernoulli variable $\zeta \sim \mathrm{Bernoulli}(\rho)$, where $\rho \in [0, 1]$ controls the probability of applying mixing.

2. If $\zeta = 1$, draw $\lambda \sim \mathcal{U}[0, 1]$ and mix activations between clean and adversarial inputs at selected classifier layers. Let $\phi_\ell(\cdot)$ denote the activation at layer $\ell$ within the classifier $g_s$. For each layer $\ell$ chosen for mixing:

$$\tilde{\phi}_\ell = (1 - \lambda)\, \phi_\ell\big(t(x + \delta)\big) + \lambda\, \phi_\ell\big(t(x)\big), \tag{10}$$

while other layers use the adversarial activations. If $\zeta = 0$, no mixing is applied and the classifier uses adversarial activations only.

The classifier then produces logits based on the mixed representation. We average the loss over transformations and stochastic draws:

$$\mathcal{J}_{\mathrm{Mix}}(x, \delta) = \mathbb{E}_{t \sim \Pi}\, \mathbb{E}_{\lambda, \zeta}\Big[\mathcal{L}\big(g_s^{\mathrm{mixed}}(t(x), t(x + \delta)),\, \tau\big)\Big], \tag{11}$$

where $g_s^{\mathrm{mixed}}$ denotes the classifier forward pass with layer-wise mixing applied, and $\tau$ is the target class label.

In our robust formulation, classifier mismatch is modeled by the Jacobian perturbation $\Delta J$ and logit offset $b$ via the term $\Delta J(h_s(x + \delta) - h_s(x)) + b$ in Eq. 7. By mixing clean and adversarial features, we shrink the effective displacement $h_s(x + \delta) - h_s(x)$ seen by the classifier, thereby reducing sensitivity to $\Delta J$ and $b$ and addressing discrepancy D2. Section 5 shows that this intuition yields a quantitative bound on the worst-case effect of $(\Delta J, b)$.

### 4.4 FEATURE ALIGNMENT

Feature Alignment mitigates decision-boundary mismatch by steering adversarial layer representations within the classifier toward those of a representative exemplar from the target class. This strategy drives adversarial examples toward semantically meaningful and architecture-consistent regions and thereby improves transferability across models while reducing reliance on the surrogate's exact decision boundaries.

Assume we have access to one sample $z^*$ from the target class $\tau$. Let $\phi_\ell(\cdot)$ denote the activation at layer $\ell$ within the surrogate classifier $g_s$. For a transformed input $t(x)$ and perturbed input $t(x + \delta)$, define:

$$\Delta_\ell^t(x, \delta) = \phi_\ell\big(t(x + \delta)\big) - \phi_\ell\big(t(x)\big), \quad d_\ell^t = \phi_\ell\big(t(z^*)\big) - \phi_\ell\big(t(x)\big), \quad v_\ell^t = \frac{d_\ell^t}{\|d_\ell^t\|_2}. \tag{12}$$

Here, $\Delta_\ell^t(x, \delta)$ is the adversarial displacement at layer $\ell$ and $v_\ell^t$ is the unit direction toward the exemplar. We encourage alignment by maximizing the projection of $\Delta_\ell^t(x, \delta)$ onto $v_\ell^t$:

$$\mathcal{L}_{\mathrm{FA}}(t; x, \delta) = \big\langle \Delta_\ell^t(x, \delta),\, v_\ell^t \big\rangle, \tag{13}$$

which drives adversarial features toward directions associated with the target class, improving cross-model transferability.

In the robust formulation, decision-boundary mismatch appears as the margin bias $\beta$ in Eq. 5, which can shift the target margin even if logits align. Encouraging progress along $v_\ell^t$ promotes local increases in the target-class margin in a direction that is consistent across architectures, thereby compensating for such boundary shifts and addressing discrepancy D3. Section 5 formalizes this via a directional monotonicity condition.

### 4.5 TART OBJECTIVE

We unify the components into a single objective that optimizes over the surrogate path $(h_s, g_s)$. For a perturbation $\delta$ with $\|\delta\| \le \varepsilon$, we maximize:

$$\max_{\|\delta\| \le \varepsilon} \underbrace{\mathbb{E}_{t \sim \Pi}\, \mathbb{E}_{\lambda, \zeta}\Big[\mathcal{L}\big(g_s^{\mathrm{mixed}}(t(x), t(x + \delta); \lambda, \zeta, \tau)\big)\Big]}_{\text{EoT + Latent Mixing}} + \underbrace{\gamma_{\mathrm{FA}}\, \mathbb{E}_{t \sim \Pi}\Big[\mathcal{L}_{\mathrm{FA}}(t; x, \delta)\Big]}_{\text{Feature Alignment}}. \tag{14}$$

Here $g_s^{\text{mixed}}$ denotes the classifier forward pass with layer-wise mixing (Eq. 10), $(\lambda, \zeta)$ control mixing, and $\mathcal{L}_{\text{FA}}$ is from Eq. 13. The first term encourages robustness under input transformations and classifier sensitivity variations, while the second biases adversarial features toward the target-class direction. We optimize Eq. 14 with I-FGSM, sampling transformations and mixing variables each iteration.

In the next section, we analyze a margin-based surrogate of Eq. 14 and show that, under local smoothness and directionality assumptions, it lower-bounds the robust targeted margin $\text{RobMarg}(x, \delta; \rho)$ defined in Eq. 7. This establishes TART as a principled relaxation of the ideal robust objective in Eq. 8.

# 5 THEORETICAL ANALYSIS: GUARANTEES UNDER LOCAL ASSUMPTIONS

We now show that a margin-based counterpart of the TART objective provides a provable lower bound on the *robust targeted margin* under mild, locally standard smoothness conditions. This formalizes TART as a principled relaxation of the ideal robust optimization objective (Section 3.3). We present the main statements and intuition here; complete proofs are given in Appendix B.

**Standing assumptions.** Within a local neighborhood $\mathcal{N}(x)$, we assume:

(A1) Feature extractors $h_s, h_t$ are $L_h$-Lipschitz.

(A2) Classifiers $g_s, g_t$ are $L_g$-Lipschitz and differentiable.

(A3) Margins $m_\tau(u) = u_\tau - \max_{k \neq \tau} u_k$ satisfy $|m_\tau(u) - m_\tau(v)| \leq \|u - v\|_\infty$.

These control feature and logit variation under small changes. An additional assumption (A4) regarding directional monotonicity will be introduced for the alignment term.

For each transformation $t \sim \Pi$, we write $z_a^t = h_s(t(x + \delta))$ and $z_0^t = h_s(t(x))$, and denote by $z_\lambda^t$ the mixed feature obtained by applying Eq. 10 along the path from $z_0^t$ to $z_a^t$. We also write $\Delta^t(x, \delta)$ for the layer-wise displacement defined in Eq. 12.

## 5.1 AUXILIARY LEMMAS

**Lemma 1: Margins are $1$-Lipschitz.** For any logits $u, v \in \mathbb{R}^C$, $|m_\tau(u) - m_\tau(v)| \leq \|u - v\|_\infty$.

**Lemma 2: Propagation through classifier.** For $z, z' \in \mathbb{R}^p$,

$$\|g_s(z) - g_s(z')\|_\infty \leq L_g \|z - z'\|_2 \quad \Rightarrow \quad |m_\tau(g_s(z)) - m_\tau(g_s(z'))| \leq L_g \|z - z'\|_2.$$

This ensures small feature differences yield bounded margin variation.

**Lemma 3: EoT coupling under extractor perturbations.** If $h_s(t(x)) = h_s(x) + \Delta_h^t + \varepsilon_t$ with $\|\Delta_h^t\| \leq \rho_h^t$ and $\mathbb{E}_t[\rho_h^t] \leq \bar{\rho}_h$, $\mathbb{E}_t\|\varepsilon_t\| \leq \bar{\epsilon}_T$, then:

$$\inf_{\|\Delta_h\| \leq \rho_h} m_\tau\big(g_s(h_s(x) + \Delta_h)\big) \geq \mathbb{E}_{t \sim \Pi}[m_\tau(g_s(h_s(t(x))))] - L_g(\rho_h + \bar{\rho}_h + \bar{\epsilon}_T). \quad (15)$$

This shows the EoT term lower-bounds a worst-case feature deviation up to a Lipschitz penalty.

**Lemma 4: Mixing mitigates Jacobian mismatch.** For $z_a^t = h_s(t(x + \delta))$, $z_0^t = h_s(t(x))$, and mixed $z_\lambda^t$:

$$\mathbb{E}_{\zeta, \lambda}\big[m_\tau(g_s(z_\lambda^t) + \Delta J(z_\lambda^t - z_0^t) + b)\big] \geq \mathbb{E}_{\zeta, \lambda}[m_\tau(g_s(z_\lambda^t))] - \rho_J \psi_{\text{mix}} \|z_a^t - z_0^t\| - \rho_b, \quad (16)$$

where $\psi_{\text{mix}} = 1 - \rho\bar{\lambda}$ captures effective displacement under stochastic mixing. This bounds Jacobian/logit mismatch penalties.

**Lemma 5: Directional lemma for alignment.** Assume directional monotonicity along the exemplar direction:

$$(A4) \qquad \langle \nabla_z m_\tau(g_s(z)), v_\ell^t \rangle \geq \kappa^t, \quad \|\nabla^2 m_\tau(g_s(z))\| \leq L_m. \quad (17)$$

Then

$$m_\tau(g_s(h_s(t(x + \delta)))) \geq m_\tau(g_s(h_s(t(x)))) + \kappa^t \langle \Delta^t(x, \delta), v_\ell^t \rangle - \frac{L_m}{2} \|\Delta^t(x, \delta)\|^2. \quad (18)$$

## 5.2 Robust Margin Guarantees

We now state our main guarantee, showing that the TART objective lower-bounds the robust targeted margin defined in Eq. 7.

**Theorem 1** (TART lower-bounds the robust targeted margin). *Under (A1)–(A4) and couplings above, define:*

$$\text{TART}_{\text{proj}}(x, \delta) = \mathbb{E}_{t,\zeta,\lambda}[m_\tau(g_s(z_\lambda^t))] + \gamma_{\text{FA}}\mathbb{E}_t[\langle \Delta^t(x, \delta), v_\ell^t \rangle]. \tag{19}$$

*Then for any $\|\delta\| \leq \varepsilon$,*

$$\text{RobMarg}(x, \delta; \rho) \geq \text{TART}_{\text{proj}}(x, \delta) - L_g(\rho_h + \bar{\rho}_h + \bar{\epsilon}_T) - \rho_J\psi_{\text{mix}}\mathbb{E}_t\|z_a^t - z_0^t\| \tag{20}$$

$$- (\rho_b + \rho_\beta) - \left(\frac{L_m}{2} - \kappa_{\min}\gamma_{\text{FA}}\right)\mathbb{E}_t\|\Delta^t(x, \delta)\|^2,$$

*where $\psi_{\text{mix}} = 1 - \rho\bar{\lambda}$ and $\kappa_{\min} = \inf_t \kappa^t$. Choosing $\gamma_{\text{FA}} \leq L_m/(2\kappa_{\min})$ makes the last term nonpositive.*

The robust margin exceeds the TART objective up to additive penalties from extractor variation, Jacobian mismatch, logit/boundary shifts, and curvature; smaller uncertainty radii tighten this gap.

# 6 Experimental Results

## 6.1 Experimental Setting

**Datasets.** Following Byun et al. (2023), we evaluate our method on the ImageNet-Compatible dataset[1] and CIFAR-10 (Krizhevsky, 2009). The ImageNet-Compatible dataset consists of 1,000 images with predefined target labels for benchmarking targeted attacks; hereafter, we refer to it as *ImageNet* for brevity. For CIFAR-10, each image is assigned a randomly selected target label different from its ground truth.

**Models.** *Target models:* CNNs (ResNet-18, WideResNet-101, BiT-M-R50, BiT-M-R101 (He et al., 2016; Zagoruyko & Komodakis, 2016; Kolesnikov et al., 2020)) and vision transformers (ViT-Base, DeiT-Base, Swin-Base, Swin-Small (Dosovitskiy et al., 2020; Touvron et al., 2021; Liu et al., 2021)). *Surrogates:* Following the attack baselines (see below), we select ResNet-50 and ViT-Tiny to ensure fair comparison. All models are initialized with pretrained weights from the PyTorch Image Models (timm) library (Wightman, 2019).

**Attack Baselines.** We compare with strong targeted transfer-attack methods, including input-transformation approaches (RDI (Zou et al., 2020), ODI (Byun et al., 2022), SU (Wei et al., 2023)) and feature-level approaches (CFM (Byun et al., 2023), FTM (Liang et al., 2025)). All baselines adopt MI (Dong et al., 2018) and TI (Dong et al., 2019) with logit loss by default. For fairness, we use the best-performing variants reported: SU with DI (Xie et al., 2019), and CFM/FTM with RDI. Official implementations and default hyperparameters are used unless noted otherwise.

**Defenses.** We evaluate against adversarial training Madry et al. (2017); Tramèr et al. (2017) and input-transformation defenses: JPEG compression Guo et al. (2017), Randomized Resizing and Padding (R&P) Xie et al. (2017), Bit Depth Reduction (Bit-R) Xu et al. (2017), and Feature Distillation (FD) Liu et al. (2019).

**Performance Metrics.** We measure attack effectiveness using the *Transfer Success Rate (TSR)*, defined as the proportion of adversarial examples that fool the target model, *conditioned on successfully attacking the surrogate(s)*. Results are averaged over 1,000 ImageNet images and the CIFAR-10 test set. Since TART achieves 100% *Generation Success Rate (GSR)* on all surrogates for both datasets, GSR is omitted.

**Implementation Details.** Following Byun et al. (2022; 2023); Liang et al. (2025), all attacks are targeted under an $L_\infty$ perturbation bound $\epsilon = 16/255$, for $T = 300$ iterations with a step size $\alpha = 2/255$. For TART, we apply feature alignment and latent mixing to the last 80% of surrogate layers (weight $\gamma_{FA} = 1$, mixing probability $\rho = 0.1$); the EoT setting uses $N = 10$ sampled transformation pairs, and exemplars are randomly drawn from the ImageNet validation set. All experiments are implemented in PyTorch and conducted on two NVIDIA RTX 3090 GPUs.

---

[1] https://github.com/cleverhans-lab/cleverhans/tree/master/cleverhans_v3.1.0/examples/nips17_adversarial_competition/dataset

Table 1: TSR (%) on CIFAR-10 and ImageNet (surrogate: ResNet-50). Bold indicates the best result.

| Dataset | Attack | CNN | | | | | ViT | | | | |
|---------|--------|-------|---------|--------|---------|------|-------|--------|--------|--------|------|
| | | RN-18 | WRN-101 | BiT-50 | BiT-101 | Avg. | ViT-B | Deit-B | Swin-B | Swin-S | Avg. |
| CIFAR-10 | RDI | 62.4 | 26.5 | 23.9 | 23.2 | 34.0 | 18.5 | 32.2 | 40.6 | 54.4 | 36.4 |
| | SU-DI | 53.7 | 22.0 | 20.2 | 16.8 | 28.2 | 12.3 | 21.7 | 31.4 | 43.1 | 27.1 |
| | ODI | 84.5 | 41.1 | 43.0 | 37.0 | 51.4 | 36.1 | 56.3 | 64.0 | 77.8 | 58.6 |
| | CFM-RDI | 94.0 | 77.6 | 62.5 | 56.8 | 72.7 | 57.7 | 74.4 | 83.3 | 90.8 | 76.6 |
| | FTM-RDI | 94.0 | 76.9 | 65.4 | 61.3 | 74.4 | 58.9 | 75.6 | 86.1 | 90.9 | 77.9 |
| | TART (ours) | **99.1** | **83.0** | **77.5** | **68.6** | **82.1** | **81.6** | **93.0** | **95.4** | **97.9** | **92.0** |
| ImageNet | RDI | 47.2 | 18.6 | 12.3 | 11.3 | 22.4 | 4.6 | 5.6 | 2.3 | 2.7 | 3.8 |
| | SU-DI | 38.6 | 15.1 | 13.5 | 8.1 | 18.8 | 3.3 | 3.2 | 1.2 | 1.6 | 2.3 |
| | ODI | 27.5 | 32.6 | 31.9 | 23.2 | 28.8 | 9.0 | 13.5 | 5.1 | 12.9 | 10.1 |
| | CFM-RDI | 83.2 | 63.7 | 56.8 | 46.1 | 62.5 | 32.0 | 30.9 | 17.2 | 21.3 | 25.4 |
| | FTM-RDI | 82.9 | 64.6 | 58.2 | 49.8 | 63.9 | 31.9 | 32.7 | 17.3 | 23.0 | 26.2 |
| | TART (ours) | **94.7** | **88.2** | **84.0** | **75.4** | **85.6** | **72.7** | **70.5** | **57.7** | **65.7** | **66.7** |

Table 2: TSR (%) on ImageNet (surrogate: ViT-Tiny). Bold indicates the best result.

| Attack | CNN | | | | | ViT | | | | |
|--------|-------|---------|--------|---------|------|-------|--------|--------|--------|------|
| | RN-18 | WRN-101 | BiT-50 | BiT-101 | Avg. | ViT-B | Deit-B | Swin-B | Swin-S | Avg. |
| RDI | 12.6 | 5.8 | 17.4 | 12.0 | 12.0 | 20.5 | 20.3 | 2.8 | 5.8 | 12.4 |
| SU-DI | 10.6 | 3.9 | 10.4 | 6.3 | 7.8 | 6.7 | 9.3 | 1.1 | 1.6 | 4.7 |
| ODI | 26.2 | 20.0 | 35.1 | 28.8 | 27.5 | 32.6 | 33.9 | 6.9 | 18.0 | 22.9 |
| CFM-RDI | 51.0 | 37.4 | 55.8 | 49.2 | 48.4 | 60.2 | 61.1 | 19.0 | 26.5 | 41.7 |
| FTM-RDI | 52.1 | 40.0 | 56.5 | 50.8 | 49.9 | 61.7 | 63.3 | 21.3 | 32.1 | 44.6 |
| TART (ours) | **82.5** | **67.7** | **86.4** | **81.4** | **79.5** | **85.5** | **87.5** | **50.7** | **63.4** | **71.8** |

Table 3: TSR (%) on ImageNet for TART with surrogate ensemble: ResNet-50 and ViT-Tiny.

| Surrogate | CNN | | | | | ViT | | | | |
|-----------|-------|---------|--------|---------|------|-------|--------|--------|--------|------|
| | RN-18 | WRN-101 | BiT-50 | BiT-101 | Avg. | ViT-B | DeiT- B | Swin-B | Swin-S | Avg. |
| RN-50 | 94.7 | 88.2 | 84.0 | 75.4 | 85.6 | 72.7 | 70.5 | 57.7 | 65.7 | 66.7 |
| ViT-T | 82.5 | 67.7 | 86.4 | 81.4 | 79.5 | 85.5 | 87.5 | 50.7 | 63.4 | 71.8 |
| RN-50 + ViT-T | 97.7 | 95.3 | 96.4 | 93.4 | 95.7 | 93.7 | 94.4 | 79.7 | 86.5 | 88.6 |

## 6.2 ADVERSARIAL TRANSFERABILITY

### 6.2.1 ATTACK ON STANDARD TARGET MODELS

**Single Surrogate.** Table 1 reports results on CIFAR-10 and ImageNet using ResNet-50 as the surrogate, while Table 2 presents ImageNet results with ViT-Tiny as the surrogate. TART outperforms all baselines across every target model. With a CNN surrogate (ResNet-50), the average TSR improvement over the strongest baseline is 7.7% on CNN targets and 14.1% on ViT targets for CIFAR-10, and 21.7% on CNN targets and 40.5% on ViT targets for ImageNet. The largest improvement occurs on ImageNet when transferring to Swin-B, where TART exceeds the best baseline by 42.7%. With a ViT surrogate (ViT-Tiny), TART achieves average TSR gains of 29.6% on CNN targets and 27.2% on ViT targets for ImageNet.

**CNN + ViT Surrogates.** Table 3 shows ImageNet results using an ensemble of ResNet-50 and ViT-Tiny. The ensemble improves TSR over the single CNN surrogate by 10.1% on CNN targets and 21.9% on ViT targets, and over the single ViT surrogate by 16.2% and 16.8%, respectively, demonstrating the benefit of diverse surrogates for cross-architecture transfer.

### 6.2.2 ATTACK ON DEFENDED TARGET MODELS

We evaluate TSR on defended targets, including three adversarially trained models and those protected by input-transformation defenses. The results are presented in Table 4. TART achieves the highest TSR under all defenses, averaging 70.2% on adversarially trained models and 71.6% on input-transformation defenses, with improvements of 39.5% and 28.7% over the strongest baselines, demonstrating its robustness against advanced defenses.

Table 4: TSR (%) on ImageNet against defenses (surrogate: ResNet-50). **Left**: adversarially trained models. **Right**: average over input-transformation defenses.

| Attack | Adversarial Training Defense | | | | Input Transformation-Based Defenses | | | | |
|---|---|---|---|---|---|---|---|---|---|
| | Inc-v3ens3 | Inc-v3ens4 | Inc-v2ens | Avg. | R&P | Bit-R | JPEG | FD | Avg. |
| RDI | 4.1 | 2.1 | 1.9 | 2.7 | 11.7 | 11.7 | 4.9 | 10.4 | 9.7 |
| SU-DI | 5.0 | 3.4 | 2.6 | 3.7 | 11.3 | 9.0 | 4.8 | 8.1 | 8.3 |
| ODI | 19.6 | 15.1 | 13.7 | 16.1 | 26.5 | 21.5 | 14.7 | 20.5 | 20.8 |
| CFM-RDI | 34.0 | 30.4 | 26.3 | 30.2 | 48.9 | 43.0 | 34.1 | 41.5 | 41.8 |
| FTM-RDI | 36.0 | 28.9 | 27.3 | 30.7 | 50.1 | 43.9 | 35.1 | 42.3 | 42.9 |
| TART (ours) | **72.3** | **68.5** | **69.8** | **70.2** | **79.8** | **75.1** | **59.6** | **72.1** | **71.6** |

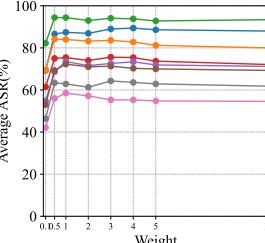 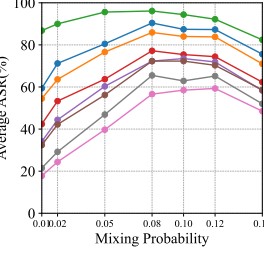 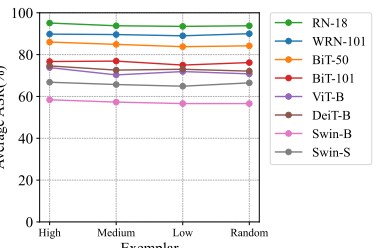

Figure 2: Weight $\gamma_{FA}$.    Figure 3: Mixing probability $\rho$.    Figure 4: Exemplar $z^*$.

Table 5: Impact of each component on ImageNet using ResNet-50 as the surrogate.

| Method Variant | CNN | | | | | ViT | | | | |
|---|---|---|---|---|---|---|---|---|---|---|
| | RN-18 | WRN-101 | BiT-50 | BiT-101 | Avg. | ViT-B | Deit-B | Swin-B | Swin-S | Avg. |
| TART w/o EoT | 81.0 | 63.8 | 57.2 | 47.9 | 62.5 | 40.9 | 39.9 | 31.0 | 35.7 | 36.9 |
| TART w/o Latent Mixing | 89.8 | 53.3 | 58.6 | 45.6 | 61.8 | 39.6 | 37.3 | 21.7 | 27.6 | 31.6 |
| TART w/o Feature Alignment | 87.8 | 82.2 | 74.4 | 68.9 | 78.3 | 66.8 | 65.1 | 52.3 | 56.6 | 60.2 |
| TART | 94.7 | 88.2 | 84.0 | 75.4 | 85.6 | 72.7 | 70.5 | 57.7 | 65.7 | 66.7 |

## 6.3 ABLATION STUDY

**Variants of TART.** We assess the contribution of each component using ImageNet with ResNet-50 as the surrogate (Table 5). Removing EoT reduces TSR by 23.1% on CNN targets and 29.8% on ViT targets. Removing latent mixing causes drops of 23.8% (CNN) and 35.1% (ViT), while excluding feature alignment leads to smaller declines of 7.3% (CNN) and 6.5% (ViT). These results confirm that EoT, latent mixing, and feature alignment are all essential for maximizing transferability.

**Hyperparameter Sensitivity.** We examine two hyperparameters: the feature-alignment weight $\gamma_{FA}$ and the mixing probability $\rho$. As shown in Figs. 2 and 3, TART achieves stable TSR for $\gamma \in [1, 5]$ and $\rho \in [0.08, 0.12]$, indicating moderate sensitivity and robust performance across these ranges.

**Effect of Exemplar Selection.** We evaluate four exemplar-selection strategies based on surrogate confidence margin: High, Medium, Low, and Random (Fig. 4). The results are similar across strategies, indicating that TART is robust to exemplar choice.

## 7 CONCLUSION

Targeted transfer attacks remain significantly more challenging than non-targeted ones due to feature-extractor mismatch, classifier sensitivity mismatch, and decision-boundary misalignment between models. We formulated targeted transfer as a robust optimization problem that maximizes the worst-case target-class margin under these uncertainties and proposed TART as a tractable relaxation. TART operationalizes this ideal objective through three components—Expectation over Transformation, Latent Mixing, and Feature Alignment with a representative exemplar—while providing a theoretical guarantee as a lower bound on the worst-case loss. Empirical results on ImageNet and CIFAR-10 demonstrate that TART achieves state-of-the-art targeted transferability across CNNs and ViTs and maintains robustness against advanced defenses. This work advances understanding of targeted transfer, enabling the development of more effective defenses against black-box adversarial attacks.

## 8 ETHICS CONSIDERATIONS

This work adheres to the ICLR Code of Ethics. Although it explores the targeted transferability of adversarial examples, it is intended to advance the understanding of model vulnerabilities and promote the development of more robust machine learning systems. All experiments use standard, publicly available datasets (ImageNet and CIFAR-10), which contain no personal or sensitive data. The research does not involve human participants or deployed systems.

## 9 REPRODUCIBILITY STATEMENT

We have made extensive efforts to ensure the reproducibility of our results. All key implementation details of the proposed method (TART), including the attack pipeline, transformation set, and optimization settings, are described in Section 6.1.

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

## A  LLM USAGE

Large Language Models (LLMs) are used solely to assist with language polishing, grammar correction, and improving the clarity of presentation.

## B  THEORETICAL ANALYSIS: GUARANTEES UNDER LOCAL ASSUMPTIONS

We show that the TART objective in Eq. equation 14 optimizes a *provable lower bound* of the robust targeted margin in Eq. equation 7 under mild local assumptions. This establishes TART as a principled relaxation of the ideal robust optimization in Section 3.3.

**Standing assumptions.**  Within a neighborhood $\mathcal{N}(x)$ we assume:

(A1) Extractors $h_s, h_t$ are $L_h$-Lipschitz.

(A2) Classifiers $g_s, g_t$ are $L_g$-Lipschitz and differentiable; the local second-order remainder of $g_s$ (when linearized at $h_s(x)$) is bounded by $L_{g,2}$ in $\ell_\infty$.

(A3) For the targeted margin $m_\tau(u) = u_\tau - \max_{k\neq\tau} u_k$, $\big|m_\tau(u) - m_\tau(v)\big| \le \|u - v\|_\infty$.

(A4) (Directional monotonicity for alignment) Along the line segment between $h_s(t(x))$ and $h_s\big(t(x + \delta)\big)$ we have $\langle \nabla_z m_\tau(g_s(z)), v_\ell^t \rangle \ge \kappa_\ell^t \ge 0$, and a local curvature bound $\|\nabla^2 m_\tau(g_s(z))\| \le L_m$.

Assumption (A4) is standard for directional lower bounding under small displacements. Unless specified, vector norms are $\ell_2$ and operator norms are spectral norms.

### B.1  AUXILIARY LEMMAS

**Lemma 1** (Margin Lipschitzness). *For any $u, v \in \mathbb{R}^C$, $\big|m_\tau(u) - m_\tau(v)\big| \le \|u - v\|_\infty$.*

**Lemma 2** (Propagation through $g_s$). *For any $z, z' \in \mathbb{R}^p$, $\|g_s(z) - g_s(z')\|_\infty \le L_g\|z - z'\|_2$ and hence $\big|m_\tau(g_s(z)) - m_\tau(g_s(z'))\big| \le L_g\|z - z'\|_2$ by Lemma 1.*

*Proof.*  $g_s$ is $L_g$-Lipschitz in $\ell_2$. Combine with Lemma 1. □

**Lemma 3** (EoT coupling under extractor variability). *Let $t = t_2 \circ t_1$ with $t_1, t_2$ sampled uniformly from $\mathcal{T}$ and $\Pi$ the induced distribution on composed transformations. Suppose there exists a coupling such that*

$$h_s(t(x)) = h_s(x) + \Delta_h^t + \varepsilon_t, \qquad \|\Delta_h^t\| \le \rho_h^t, \quad \mathbb{E}_{t\sim\Pi}[\rho_h^t] \le \bar{\rho}_h, \quad \mathbb{E}_{t\sim\Pi}\|\varepsilon_t\| \le \bar{\epsilon}_T.$$

*Then for any $\rho_h \ge 0$,*

$$\inf_{\|\Delta_h\| \le \rho_h} m_\tau\big(g_s(h_s(x) + \Delta_h)\big) \ge \mathbb{E}_{t\sim\Pi}\Big[m_\tau\big(g_s(h_s(t(x)))\big)\Big] - L_g\Big(\rho_h + \bar{\rho}_h + \bar{\epsilon}_T\Big). \tag{21}$$

*Proof.*  For any feasible $\Delta_h$ with $\|\Delta_h\| \le \rho_h$ and any $t$,

$$\|h_s(x) + \Delta_h - h_s(t(x))\| = \|h_s(x) + \Delta_h - (h_s(x) + \Delta_h^t + \varepsilon_t)\|$$
$$= \|\Delta_h - \Delta_h^t - \varepsilon_t\|$$
$$\le \|\Delta_h - \Delta_h^t\| + \|\varepsilon_t\|$$
$$\le \|\Delta_h\| + \|\Delta_h^t\| + \|\varepsilon_t\|$$
$$\le \rho_h + \rho_h^t + \|\varepsilon_t\|.$$

By Lemma 2,

$$m_\tau\big(g_s(h_s(x) + \Delta_h)\big) \ge m_\tau\big(g_s(h_s(t(x)))\big) - L_g\big(\rho_h + \rho_h^t + \|\varepsilon_t\|\big).$$

Taking the expectation over $t \sim \Pi$ and then the infimum over $\|\Delta_h\| \le \rho_h$ yields

$$\inf_{\|\Delta_h\| \le \rho_h} m_\tau\big(g_s(h_s(x) + \Delta_h)\big) \ge \mathbb{E}_{t\sim\Pi}\Big[m_\tau\big(g_s(h_s(t(x)))\big)\Big] - L_g\Big(\rho_h + \mathbb{E}_{t\sim\Pi}[\rho_h^t] + \mathbb{E}_{t\sim\Pi}\|\varepsilon_t\|\Big),$$

and using $\mathbb{E}_t[\rho_h^t] \le \bar{\rho}_h$ and $\mathbb{E}_t\|\varepsilon_t\| \le \bar{\epsilon}_T$ gives Eq. equation 21. □

**Lemma 4** (Layer-wise mixing controls Jacobian mismatch). *Let $z_a^t = h_s\big(t(x + \delta)\big)$ and $z_0^t = h_s\big(t(x)\big)$. In $g_s$, perform layer-wise mixing as in Eq. equation 10 with $\zeta \sim \text{Bernoulli}(\rho)$ and $\lambda \sim \mathcal{U}[0, 1]$ when $\zeta = 1$, and no mixing ($\zeta = 0$) otherwise. Let the effective feature input to $g_s$ be $z_\lambda^t$ (the output of the last mixed layer). Then for any $\|\Delta J\| \leq \rho_J$ and $\|b\|_\infty \leq \rho_b$,*

$$\mathbb{E}_{\zeta,\lambda}\, m_\tau\big(g_s(z_\lambda^t) + \Delta J(z_\lambda^t - z_0^t) + b\big) \;\geq\; \mathbb{E}_{\zeta,\lambda}\, m_\tau\big(g_s(z_\lambda^t)\big) \;-\; \rho_J\, \psi_{\text{mix}} \|z_a^t - z_0^t\| \;-\; \rho_b, \quad (22)$$

*where*

$$\psi_{\text{mix}} \;=\; \mathbb{E}_{\zeta,\lambda}\, \frac{\|z_\lambda^t - z_0^t\|}{\|z_a^t - z_0^t\|} \;=\; (1 - \rho) \cdot 1 \;+\; \rho\, \mathbb{E}[1 - \lambda] \;=\; 1 - \rho\, \bar{\lambda}, \quad \bar{\lambda} = \mathbb{E}[\lambda] = \tfrac{1}{2}.$$

*Proof.* By the mixing rule, when $\zeta = 1$ we obtain $z_\lambda^t = (1 - \lambda) z_a^t + \lambda z_0^t$ at the classifier input (or its equivalent feature handed to $g_s$ after the mixed stack), hence $\|z_\lambda^t - z_0^t\| = (1 - \lambda)\|z_a^t - z_0^t\|$. When $\zeta = 0$, $z_\lambda^t = z_a^t$ so the factor is 1. Therefore the expected scaling factor is $\psi_{\text{mix}} = (1 - \rho) \cdot 1 + \rho\, \mathbb{E}[1 - \lambda] = 1 - \rho\bar{\lambda}$. Next, apply margin Lipschitzness (Lemma 1) with the $\ell_\infty$-norm and the bound $\|\Delta J(z_\lambda^t - z_0^t)\|_\infty \leq \|\Delta J\|\, \|z_\lambda^t - z_0^t\|$, to get the inequality after taking expectation over $\zeta, \lambda$ and subtracting the bounded logit offset $\rho_b$. $\qquad\square$

**Lemma 5** (Directional lower bound for projection alignment). *For transformation $t$, define the adversarial displacement at a reference layer (or aggregated across the mixed stack) by $\Delta^t(x, \delta) = \phi_\ell\big(t(x + \delta)\big) - \phi_\ell\big(t(x)\big)$ and the exemplar direction $v_\ell^t = d_\ell^t / \|d_\ell^t\|_2$ with $d_\ell^t = \phi_\ell\big(t(z^*)\big) - \phi_\ell\big(t(x)\big)$. Under (A4),*

$$m_\tau\big(g_s(h_s(t(x + \delta)))\big) \geq m_\tau\big(g_s(h_s(t(x)))\big) + \sum_\ell w_\ell\, \kappa_\ell^t\, \langle\Delta^t(x, \delta), v_\ell^t\rangle - \frac{L_m}{2} \sum_\ell w_\ell \|\Delta^t(x, \delta)\|^2,$$

$$(23)$$

*for nonnegative weights $w_\ell$.*

*Proof.* Along the line segment $z(s) = \phi_\ell\big(t(x)\big) + s\, \Delta^t(x, \delta)$,

$$m_\tau(g_s(z(1))) = m_\tau(g_s(z(0))) + \int_0^1 \langle\nabla_z m_\tau(g_s(z(s))), \Delta^t\rangle\, ds.$$

Decompose $\Delta^t$ into the exemplar direction $v_\ell^t$ and the orthogonal component. By (A4), the directional derivative along $v_\ell^t$ is at least $\kappa_\ell^t$, and the transverse component is controlled by curvature $L_m$. Summing over $\ell$ with weights $w_\ell$ yields Eq. equation 23. $\qquad\square$

### B.2 MAIN THEOREM: TART LOWER-BOUNDS THE ROBUST TARGETED MARGIN

**Theorem 1** (TART lower-bounds the robust targeted margin). *Let $\text{RobMarg}(x, \delta; \rho)$ be defined in Eq. equation 7 with the composite uncertainty set in Eq. equation 6. Consider the TART objective in Eq. equation 14, where EoT uses the transformation composition $t = t_2 \circ t_1$, latent mixing follows Eq. equation 10 with $\zeta \sim \text{Bernoulli}(\rho)$ and $\lambda \sim \mathcal{U}[0, 1]$, and Feature Alignment uses the projection loss in Eq. equation 13. Define*

$$\text{TART}_{\text{proj}}(x, \delta) = \mathbb{E}_{t \sim \Pi}\, \mathbb{E}_{\zeta,\lambda}\Big[m_\tau\big(g_s^{\text{mixed}}(t(x), t(x + \delta); \lambda, \zeta)\big)\Big] \quad (24)$$

$$+ \gamma_{\text{FA}}\, \mathbb{E}_{t \sim \Pi}\Big[\sum_\ell w_\ell\langle\Delta^t(x, \delta), v_\ell^t\rangle\Big].$$

*Then, under (A1)–(A4),*

$$\text{RobMarg}(x, \delta; \rho) \;\geq\; \text{TART}_{\text{proj}}(x, \delta) - \underbrace{L_g\Big(\rho_h + \bar{\rho}_h + \bar{\epsilon}_T\Big)}_{\textit{EoT penalty}} \quad (25)$$

$$- \underbrace{\rho_J\, \psi_{\text{mix}}\, \mathbb{E}_{t \sim \Pi}\|h_s(t(x + \delta)) - h_s(t(x))\|}_{\textit{Jacobian penalty}}$$

$$- (\rho_b + \rho_\beta) - \underbrace{\Big(\tfrac{L_m}{2} - \underline{\kappa}\, \gamma_{\text{FA}}\Big) \mathbb{E}_{t \sim \Pi} \sum_\ell w_\ell \|\Delta^t(x, \delta)\|^2}_{\textit{curvature term (non-positive if } \gamma_{\text{FA}} \leq L_m/(2\underline{\kappa}))},$$

*where $\psi_{\text{mix}} = 1 - \rho\,\bar{\lambda}$ with $\bar{\lambda} = \mathbb{E}[\lambda] = \tfrac{1}{2}$, and $\underline{\kappa} = \inf_{t, \ell} \kappa_\ell^t$.*

*Proof.* **Step 1: Start from the robust margin.** By Eq. equation 7,

$$\mathsf{RobMarg}(x, \delta; \rho) = \inf_{\substack{\|\Delta_h\| \le \rho_h, \|\Delta J\| \le \rho_J, \\ \|b\|_\infty \le \rho_b, |\beta| \le \rho_\beta}} \big\{ m_\tau\big(g_s(h_s(x+\delta) + \Delta_h) + \Delta J(z_a - z_0) + b\big) - \beta \big\},$$

where $z_a = h_s(x+\delta)$ and $z_0 = h_s(x)$.

**Step 2: Control extractor uncertainty via EoT (Lemma 3).** Apply Eq. equation 21 at $x+\delta$:

$$\inf_{\|\Delta_h\| \le \rho_h} m_\tau\big(g_s(h_s(x+\delta) + \Delta_h)\big) \ge \mathbb{E}_{t \sim \Pi} m_\tau\big(g_s(h_s(t(x+\delta)))\big) - L_g\big(\rho_h + \bar{\rho}_h + \bar{\epsilon}_T\big).$$

**Step 3: Control Jacobian/logit mismatch via layer-wise mixing (Lemma 4).** For each $t$, introduce layer-wise mixing inside $g_s$ to obtain the mixed input $z_\lambda^t$ and use Eq. equation 22:

$$\inf_{\|\Delta J\| \le \rho_J, \|b\|_\infty \le \rho_b} \mathbb{E}_{\zeta, \lambda} m_\tau\big(g_s(z_\lambda^t) + \Delta J(z_\lambda^t - z_0^t) + b\big) \ge \mathbb{E}_{\zeta, \lambda} m_\tau\big(g_s(z_\lambda^t)\big) - \rho_J \psi_{\mathrm{mix}} \|z_a^t - z_0^t\| - \rho_b.$$

Averaging over $t$ gives the second penalty term in Eq. equation 25.

**Step 4: Account for boundary bias.** Subtract the worst-case boundary offset $\rho_\beta$ from the margin (by Eq. equation 5).

**Step 5: Add projection alignment and control curvature (Lemma 5).** By Lemma 5, for each $t$,

$$m_\tau\big(g_s(h_s(t(x+\delta)))\big) \ge m_\tau\big(g_s(h_s(t(x)))\big) + \sum_\ell w_\ell \, \kappa_\ell^t \, \langle \Delta^t, v_\ell^t \rangle - \frac{L_m}{2} \sum_\ell w_\ell \|\Delta^t\|^2.$$

Adding the explicit projection term $\gamma_{\mathrm{FA}} \sum_\ell w_\ell \langle \Delta^t, u_\ell^t \rangle$ to the objective preserves a lower bound as long as the quadratic curvature term is nonpositive; this is ensured by choosing $\gamma_{\mathrm{FA}} \le L_m/(2\underline{\kappa})$, yielding the last (nonpositive) term in Eq. equation 25. Taking expectations over $t$ and $\zeta, \lambda$, and collecting all penalties finishes the proof. $\qquad\square$

**Interpretation.** Eq. equation 25 shows that maximizing $\mathsf{TART}_{\mathrm{proj}}$ improves a *conservative* estimate of the robust margin. Each discrepancy contributes an explicit, interpretable penalty: (i) the EoT penalty shrinks as the transformation ensemble better matches extractor variability; (ii) the Jacobian penalty scales with the expected effective displacement factor $\psi_{\mathrm{mix}} = 1 - \rho\bar{\lambda}$; (iii) boundary/logit shifts add $\rho_\beta + \rho_b$; (iv) the curvature term vanishes when $\gamma_{\mathrm{FA}} \le L_m/(2\underline{\kappa})$. This explains why EoT, layer-wise mixing, and projection alignment jointly improve targeted transfer.

