# OpenReview forum: "Towards Reliable Transferability of Targeted Adversarial Attacks against Model Discrepancy"
_ICLR.cc/2026/Conference — Submitted to ICLR 2026_

### Official Review · Reviewer_VgAa · 2025-10-26

**Soundness:** 2
**Presentation:** 3
**Contribution:** 2
**Rating:** 4
**Confidence:** 4

**Summary:**

This paper studies the problem of targeted transferability of adversarial examples across heterogeneous models. It formulates a unified robust optimization framework that jointly considers discrepancies in feature extractors, classifier Jacobians, and decision boundaries. Building on this formulation, the authors propose TART, an attack method that integrates Expectation-over-Transformation (EoT), Latent Mixing, and Feature Alignment to enhance targeted transferability. Under several assumptions, they provide a theoretical analysis showing that TART optimizes a lower bound of the robust margin. Experiments on standard benchmarks demonstrate the competitiveness of TART compared to existing transfer-based attacks.

**Strengths:**

1. The paper is written in a fluent and accessible manner, making even technically involved concepts easy to follow. The overall organization is logical and reader-friendly.
2. The authors propose a relatively general formulation that explicitly models multiple sources of discrepancy between surrogate and target models (feature, Jacobian, and decision boundary), and then design algorithmic modules tailored to each. This gives a clear sense of completeness and conceptual coherence.
3. The method is supported by a theoretical framework that, under certain assumptions, provides partial guarantees relating the proposed TART objective to a lower bound of the robust targeted margin.

**Weaknesses:**

Majors:
1. The defense baselines ("Defenses" in Section 6.1) are mostly on early (many from 2017), which are no longer representative of the current state of adversarial robustness research. Including more recent baselines would make the evaluation more convincing.
2. The paper would benefit from a schematic figure illustrating the formalization in Section 3 and the method in Section 4. A visual depiction of the interactions among the EoT, Mixing, and Feature Alignment components would significantly improve clarity.
3. The proof of Lemma 3 contains a mathematical flaw: it incorrectly upper-bounds $\Vert \Delta_h-\Delta_h^t\Vert$ by $|\rho_h-\rho_h^t|$ (Line 690, page 13), whereas the correct bound should be $\rho_h+\rho_h^t$ by the triangle inequality. This leads to an underestimated Lipschitz penalty term and thus affects the tightness of the main theorem’s lower-bound guarantee. As this lemma underpins the core theoretical claim, the issue undermines the rigor of the overall guarantee.


Minors:
1. Some places do not specify which norm ($\ell_2$, $\ell_{\infty}$, etc.) is being used; also, dimensions of key symbols should be clarified when possible.
2. The notation $\rho$ is overloaded, it denotes the uncertainty radius in Section 3 and the Bernoulli parameter in Section 4. Similarly, $\epsilon$ and $\varepsilon$ should be used consistently for different quantities.
3. Line 273 (page 6): the expression "$\delta \in \Delta$" seems incorrect since $\Delta$ is not defined.
4. Line 222 (page 5): the sentence “Using a logit loss $\mathcal{L}$ to avoid saturation." lacks a subject.
5. The appendix contains many redundant expressions like “Eq. equation”; these should be cleaned up.
6. The term Jacobian mismatch should be clarified.
7. There are some incorrect uses of \citet and \citep (e.g., "Defenses" in Section 6.1).
8. In Figure 1, the vertical axis should be labeled TSR instead of ASR.
9. It would be helpful to cite *"Toward Robust Learning via Core Feature-Aware Adversarial Training"*, IEEE TIFS 2025, which is relevant to the topic.

**Questions:**

See Weaknesses. I am willing to raise the score if the authors address my concerns.

---

> ### Author Response · Authors · 2025-12-03
>
> **W.1** The defense baselines ("Defenses" in Section 6.1) are mostly on early (many from 2017), which are no longer representative of the current state of adversarial robustness research. Including more recent baselines would make the evaluation more convincing.
>
> **A.1**
> Our defense evaluation was not intended as a comprehensive robustness benchmark, but to test whether TART’s gains survive under the _standard defenses used in recent transfer-attack work_. The “classic” defenses we include (e.g., adversarial training, input denoising) are exactly those adopted by recent transferability papers [1,2,3].
>
> To address your concern, we additionally evaluated TART against a **recent test-time purification defense** of Tang & Zhang 2024 [4], which applies FGSM-based purification before inference. As shown below, under this strong defense TART still substantially outperforms all baselines in targeted success rate:
>
> |                     | RDI  | SU-DI | ODI  | CFM-RDI | FTM-RDI | TART (Ours) |
> | ------------------- | ---- | ----- | ---- | ------- | ------- | ----------- |
> | Tang & Zhang (2024) | 11.0 | 9.5   | 15.1 | 21.2    | 21.3    | **34.6**    |
>
> These results indicate that TART’s improvements over baselines persist even under a recent, strong purification defense.
>
> [1] Tang, B., Wang, Z., Bin, Y., Dou, Q., Yang, Y., & Shen, H. T. (2024). Ensemble diversity facilitates adversarial transferability. In Proceedings of the IEEE/CVF Conference on Computer Vision and Pattern Recognition (pp. 24377-24386).
>
> [2] Wang, K., He, X., Wang, W., & Wang, X. (2024). Boosting adversarial transferability by block shuffle and rotation. In Proceedings of the IEEE/CVF conference on computer vision and pattern recognition (pp. 24336-24346).
>
> [3] Chen, B., Yin, J., Chen, S., Chen, B., & Liu, X. (2023). An adaptive model ensemble adversarial attack for boosting adversarial transferability. In Proceedings of the IEEE/CVF international conference on computer vision (pp. 4489-4498).
>
> [4] Tang, L., & Zhang, L. (2024). Robust overfitting does matter: Test-time adversarial purification with fgsm. In Proceedings of the IEEE/CVF Conference on Computer Vision and Pattern Recognition (pp. 24347-24356).
>
> **W.2** The paper would benefit from a schematic figure illustrating the formalization in Section 3 and the method in Section 4. A visual depiction of the interactions among the EoT, Mixing, and Feature Alignment components would significantly improve clarity.
>
> **A.2** Thank you for the suggestion. In the revised version, we have added a schematic method figure (Fig. 1) that explicitly illustrates the three components of TART—EoT, latent mixing, and feature alignment—and how they interact during the attack optimization.
>
> **W.3** The proof of Lemma 3 contains a mathematical flaw: it incorrectly upper-bounds $\|\Delta_h - \Delta_{h}^t\|$ by $|\rho_h - \rho_{h}^t|$ (Line 690, page 13), whereas the correct bound should be $\rho_h + \rho_{h}^t$ by the triangle inequality. This leads to an underestimated Lipschitz penalty term and thus affects the tightness of the main theorem’s lower-bound guarantee. As this lemma underpins the core theoretical claim, the issue undermines the rigor of the overall guarantee.
>
> **A.3** Thank you for carefully checking Lemma 3. You are correct that the original proof incorrectly used $\|\Delta_h - \Delta_h^{\,t}\| \le |\rho_h - \rho_h^{\,t}|$, which is a reverse–triangle-inequality lower bound, not an upper bound.
>
> In the **revised version (changes marked in blue)**, we have:
>
> - **Corrected Lemma 3 and its proof** to use the triangle-inequality bound $\|\Delta_h - \Delta_h^{\,t}\| \le \rho_h + \rho_h^{\,t}$. This leads to the EoT-related penalty term$-L_g(\rho_h + \bar\rho_h + \bar\epsilon_T)$ instead of the earlier $-L_g(|\rho_h - \bar\rho_h| + \bar\epsilon_T)$. The corresponding statement of Lemma 3 in the main text has been updated consistently.
> - **Updated the EoT-related penalty term in Theorem 1 and its proof** so that all bounds use this corrected expression.
>
> This fix makes the EoT-related Lipschitz penalty slightly larger (the bound becomes looser) but does **not** change the structure or validity of the main guarantee: under (A1)–(A4), the margin-based TART objective still lower-bounds the robust targeted margin. Our algorithm and empirical results do not rely on this term being numerically tight; the bound is used to decompose the robust margin into contributions from extractor variability, classifier mismatch, and boundary shifts, and that decomposition remains intact.

---

### Official Review · Reviewer_eoEJ · 2025-10-29

**Soundness:** 2
**Presentation:** 3
**Contribution:** 2
**Rating:** 4
**Confidence:** 5

**Summary:**

This paper first theorizes the transferability chanlleges into three mismatches: extractor mismatch, classifier mismatch, and decision boundary mismatch. Afterwards, the paper proposes TART, which consists of three key elements, Expectation over Transformation, Latent Mixing, and Feature Alignment to compensate for respective mismatch. Finally, theoretical and experimental validations are presented to demonstrate the effectiveness of TART.

**Strengths:**

1. The three mismatches, i.e., extractors, classifier, and decision booundary, are generally intuitive to capture the main challenges in transfer-based attacks.
2. The overall writing, though aided by LLM, is clear and easy to follow.

**Weaknesses:**

This paper suffers from three major weaknesses: **Theory**, **Method** and **Experiments**.

**Theory**: First of all, the three mismatches, although intuitive, are innate definitions of black-box attacks. For example, extractor mismatch (EM) is the primary prerequisite for a 'black-box' attack, i.e., victim models use a different structure than the surrogate ones. The following classifier/decision boundary mismatches (CM, DBM) are all required for similar reasons. Besides, according to the information flow within models, EM, CM and DBM are not necessarily independent, as the former always contributes to the latter. Lastly, the idea of directly resolving these mismatches is paradoxical because if there are no mismatches, it is no longer viable to consider this attack as 'black-box'.

**Method**: Second, the three components of TART, despite the claim that they compensate for the three mismatches, are all well-established methods and widely adopted for boosting transferability. i) EoT, which randomly uses a group of transformations, has been proven effective for better robustness in both earlier [1] and recent works [2]. ii) Latent mixing, which mixes clean features/activations into adversarial ones, as is done in one of the baseline CFM, is also an established method. This paper proposes to use activation in the classifier layers instead of features without clearly explaining the theoretical necessity and demonstrating the experimental superiority. iii) Feature alignment, which essentially pushes the adversarial example further away from the original ones and towards the targeted ones, exactly follows the idea of a triplet structure, i.e., using the original example as a negative, targeted one as a positive. This triplet structure has also been widely adopted in transfer-based attacks [2] and deep metric learning attacks/defenses [3]. In sum, TART falls short regarding the novelty of methodology as it combines several proven effective methods.

**Experiments**: Lastly, the experiments suffer from unexpalined inconsistency. While the overall settings follow CFM, the target models vary significantly from the CFM paper. The only few consistent setttings exhit significant performance gaps. For example, for RN-50 against RN-18 on ImageNet, CFM-RDI reports 88.4% TSR in the original paper but 83.2%, with a 5% performance gap. These inconsistency further undermines the experimental solidarity of the paper.


[1] Lu, D., Wang, Z., Wang, T., Guan, W., Gao, H., & Zheng, F. (2023). Set-level guidance attack: Boosting adversarial transferability of vision-language pre-training models. In Proceedings of the IEEE/CVF International Conference on Computer Vision (pp. 102-111).

[2] Gao, S., Jia, X., Ren, X., Tsang, I., & Guo, Q. (2024, September). Boosting transferability in vision-language attacks via diversification along the intersection region of adversarial trajectory. In European Conference on Computer Vision (pp. 442-460). Cham: Springer Nature Switzerland.

[3] Tian, Q., Lin, C., Zhao, Z., Li, Q., & Shen, C. (2024, July). Collapse-aware triplet decoupling for adversarially robust image retrieval. In Proceedings of the 41st International Conference on Machine Learning (pp. 48139-48153).

**Questions:**

Q1. Could you please clarify the superiority&necessity of using activation mixing instead of features? I notice that the mixing probability used in the paper is also the same as that in CFM (p=0.1), could you share your insight on why using feature and activation in latent mixing yields the same optimal hyperparameters?

Q2. Please clarify the variation of target models as well as the noticeable performance inconsistency of CFM-RDI regarding RN-50 against RN-18 on ImageNet.

---

> ### Author Response · Authors · 2025-12-03
>
> **W.1** Theory Weakness
>
> **A.1**
>
> 1. **We do not aim to eliminate mismatch, but to be robust to it.**
>    We fully agree that feature-, classifier-, and boundary-level mismatches are inherent to black-box transfer. Our goal is **not** to drive these discrepancies to zero (which would indeed reduce to a white-box setting), but to explicitly model them via an uncertainty set and construct perturbations whose **worst-case margin remains high for all variations within that set**, resulting in adversarial examples that are highly likely to transfer from the surrogate model to an unknown target model within this family. The radii $\rho_h, \rho_J, \rho_\beta$ are therefore **not** “targets to shrink to zero,” but parameters that describe how much mismatch the attack should tolerate.
>
> 2. **The decomposition is conceptual; we do not assume independence.**
>    EM, CM, and DBM are clearly related along the information flow, and we do not claim statistical independence. In our formulation they are collected jointly in the composite uncertainty set $\mathcal{U}(\rho)$. The decomposition into D1 (feature), D2 (classifier/Jacobian), and D3 (boundary bias) is **conceptual and analytic**: it separates different _modes_ of surrogate–target disagreement (local feature mapping, local sensitivity, residual boundary shift) so that each TART component can be tied to a specific part of the robust objective. The analysis (Lemmas 3–5, Theorem 1) works with the full composite uncertainty and does not require these terms to be independent.
>
> In short, we explicitly model mismatches that are inherent to black-box attacks and design TART to be robust to them; we do not attempt to “remove” mismatch or collapse the setting into white-box.

---

> ### Author Response · Authors · 2025-12-03
>
> **W.2** The novelty of methodology
>
> **A.2**
> We agree that each of the three components of TART has roots in prior work, and we do **not** claim them as new primitives. The novelty is how they are **derived from and tied to** our robust-optimization formulation of targeted transfer, rather than their individual existence.
>
> 1. **Principled role of the three components.**
>    In our formulation, the target model is treated as unknown and variable, and surrogate–target discrepancy is modeled via a composite uncertainty set over feature mismatch $\Delta_h$, classifier (Jacobian) mismatch $(\Delta J,b)$, and boundary bias $\beta$. In the **revised paper**, Sec. 4.1 (*“Relaxing the Robust Objective”*) makes explicit that:
>    - EoT is used as a relaxation of feature-extractor mismatch $\Delta_h$ (D1),
>    - Latent mixing is introduced to relax classifier mismatch $(\Delta J,b)$ (D2),
>    - Feature alignment is used to mitigate boundary bias $\beta$ (D3).
>
>    Sec. 5 (Lemmas 3–5 and Theorem 1) then proves that the resulting margin-based TART objective **lower-bounds the robust targeted margin** under this uncertainty set. So, while the components themselves are known, their **one-to-one mapping to the uncertainty model and the associated lower-bound analysis** are new.
>
> 2. **Latent mixing vs. prior feature mixing (e.g., CFM).**
>    D2 specifically concerns the mapping from features to logits, i.e., the classifier $g_s$. Our latent mixing therefore blends clean and adversarial activations **inside the classifier layers for the same sample**, with the explicit goal of shortening the displacement $h_s(x+\delta)-h_s(x)$ that interacts with the Jacobian mismatch term $\Delta J$ in Eq. (4).
>    By contrast, CFM mixes intermediate features **across different inputs** to empirically encourage transferability, but is not designed or analyzed as a mechanism to control classifier–Jacobian mismatch. Our mixing location (within $g_s$) and objective (reducing the effect of $\Delta J$ are chosen to match the uncertainty model and are precisely what is captured in Lemma 4.
>
> 3. **Feature alignment vs. generic triplet-style objectives.**
>    Our feature alignment term uses a **directional projection** of the adversarial displacement onto the exemplar direction, instead of a generic triplet loss. This is intentional: it matches the directional monotonicity assumption (A4) and allows us to relate the alignment term directly to the boundary bias $\beta$ and curvature in Lemma 5 and Theorem 1. Prior triplet-based attacks do not connect their loss to a robust margin under an explicit uncertainty set.
>
> 4. **Empirical evidence beyond heuristic combinations.**
>    Consistent with this structured design, TART achieves **substantial gains over existing targeted transfer attacks**, especially in the challenging heterogeneous settings (e.g., CNN $\to$ ViT and ViT $\to$ CNN), where we often observe improvements of tens of percentage points in targeted success rate. This indicates that our principled relaxation and component choices are not just theoretically motivated, but also practically more effective than prior heuristic combinations.
>
> In summary, TART does not claim novelty in EoT or feature alignment themselves; the contribution is the **robust-optimization view of targeted transfer and the derivation, analysis, and empirical validation of TART within that view**, which explains and justifies why these particular mechanisms, instantiated in this way, reduce surrogate–target discrepancy beyond existing methods.
>
> **W.3 & Q.2** Experiments
>
> **A.3** The reported gap (e.g., RN-50 → RN-18: 88.4\% TSR in CFM vs. 83.2\% in our table) comes from **model initialization differences**, not from changes to CFM itself.
>
> - In the **original CFM code**, CNNs (e.g., ResNet-18/50) are loaded from `torchvision`, while ViTs are loaded from `timm`.
> - In our experiments, as stated in Sec. 6.1, we initialize **all** surrogate and target models from the **same library** (`timm`). This ensures that every attack method (including CFM-RDI) is evaluated on exactly the same model checkpoints and preprocessing pipeline, so differences in performance reflect the attack algorithms rather than differences in pretrained weights.
>
> We do **not** modify any algorithmic detail or hyperparameters of CFM-RDI (or other baselines); we use the official released code and default settings, changing only the model loading to `timm` for consistency across all models and methods. This explains the small absolute differences from the original CFM paper, while keeping **relative comparisons** between TART and all baselines internally consistent and fair under our shared evaluation protocol.

---

> ### Author Response · Authors · 2025-12-03
>
> **Q1** Could you please clarify the superiority&necessity of using activation mixing instead of features? I notice that the mixing probability used in the paper is also the same as that in CFM (p=0.1), could you share your insight on why using feature and activation in latent mixing yields the same optimal hyperparameters?
>
> **A.4**
> 1. **Why activation mixing in the classifier instead of feature mixing in the backbone?**
>    In our uncertainty model, D2 specifically captures _classifier-level_ mismatch via $(\Delta J, b)$ in Eq. (4), i.e., how logits change with small changes in features. To relax this term in a principled way, we need to shorten the displacement **seen by the classifier**, not just by the backbone. For that reason, our latent mixing:
>
>    - blends clean and adversarial activations **inside the classifier layers for the same image**, and
>    - does so after nonlinearities, directly in the representation space where $g_s$ operates.
>
>    This is exactly what Lemma 4 analyzes: mixing along the path between clean and adversarial classifier activations reduces the norm of the displacement multiplied by $\Delta J$, and thus the worst-case penalty from classifier mismatch. In contrast, CFM mixes backbone features (often across different inputs) as feature-level augmentation to encourage transferability, but it is not designed or analyzed to specifically control the classifier–Jacobian mismatch term. Our activation mixing is therefore not just a cosmetic variant; it is placed precisely where D2 acts and is the mechanism that makes the theoretical relaxation of the $\Delta J$ term work.
>
> 2. **Why does $p = 0.1$ also work well here?**
>    The fact that we also use $p = 0.1$ is not because mixing features and mixing classifier activations are equivalent, but because both mechanisms share a qualitative trade-off:
>
>    - if $p$ is too large, the model sees mostly mixed (almost-clean) activations and the attack weakens;
>    - if $p$ is too small, mixing has little effect on robustness to mismatch.
>
>    We used $p = 0.1$ as a reasonable starting point (following CFM) and found it to work well; performance was not highly sensitive to small changes around this value.
>
> 3. **Empirical support.**
>    Beyond this theoretical motivation, TART with classifier-layer activation mixing significantly outperforms CFM on the same benchmarks, especially in heterogeneous transfers (e.g., CNN $\to$ ViT and ViT $\to$ CNN in Tables 1–3). This suggests that our specific instantiation of mixing inside the classifier is not only theoretically aligned with D2, but also practically more effective than prior feature-level mixing.

---

### Official Review · Reviewer_4VRw · 2025-10-31

**Soundness:** 3
**Presentation:** 3
**Contribution:** 3
**Rating:** 6
**Confidence:** 3

**Summary:**

This paper addresses targeted adversarial transfer attacks and proposes TART, which aims to improve transferability from a surrogate to an unknown target model. The method is based on a robust-objective formulation and integrates three components: EoT (Expectation over Transformation), latent mixing, and feature alignment toward a target-class exemplar. Empirical results demonstrate improved targeted transfer success rates, and the paper provides theoretical analysis connecting the surrogate objective to an ideal robust objective.

**Strengths:**

1 Identifies key challenges in targeted transfer attacks: feature mismatch, classifier sensitivity, and decision-boundary shifts. Formalizes an ideal robust objective and a tractable surrogate, providing a principled framework.

2 The method is intuitive and aligns well with the theoretical framework.

3 Ablations show that each component contributes positively.

**Weaknesses:**

1 The proposed approach appears incremental, combining several existing methods

2 The conditional lower-bound guarantee is interesting, but relies on standard assumptions. Its practical impact is mostly heuristic justification rather than a strict bound.

**Questions:**

1 Section 6.3 mentions exemplar choice for feature alignment, but the discussion is brief. Please clarify how sensitive TART’s performance is to different exemplars.

2 Can the authors clarify how often the directional monotonicity assumption (Assumption A4) holds in practice for high-dimensional, non-convex networks, and whether violations of this assumption affect the reliability of the surrogate objective?

---

> ### Author Response · Authors · 2025-12-03
>
> **W.1** The proposed approach appears incremental, combining several existing methods
>
> **A.1** We agree that the _building blocks_ we use (EoT, feature mixing, feature alignment) are not new, and we do not claim them as such. The concern that the method is merely an incremental combination arises from seeing these components without the context of how they are derived and analyzed.
>
> What is new in our work, **beyond what is stated in the contribution list in Section 1**, is:
>
> - We explicitly **treat the target model as unknown and variable** and model surrogate–target discrepancy via a composite uncertainty set over feature mismatch, classifier (Jacobian) mismatch, and boundary bias. This yields a concrete worst-case targeted margin objective (Eq. (7)) for _targeted transfer_—a setting where such a structured robust formulation has **not**, to our knowledge, been developed.
>
> - In the **revised paper**, Sec. 4.1 now makes explicit that TART is constructed as a _relaxation of this max–min problem_: each component is introduced because it approximates a specific part of the uncertainty model (feature, classifier, boundary), rather than as an ad hoc mix of tricks. Sec. 5 then shows, via Lemmas 3–5 and Theorem 1, that the margin-based TART objective **lower-bounds the robust targeted margin** under this uncertainty set—i.e., maximizing our practical TART objective provably improves a surrogate that is monotone with respect to the ideal robust goal, and can be used in place of the intractable max–min.
>
> In other words, our contribution is not the mechanisms themselves, but the **robust-optimization view of targeted transfer and the derivation and analysis of TART within that view**. This provides a principled explanation of why these particular mechanisms, in this configuration, reduce surrogate–target discrepancy, which goes beyond prior heuristic combinations.
>
> **W.2** The conditional lower-bound guarantee is interesting, but relies on standard assumptions. Its practical impact is mostly heuristic justification rather than a strict bound.
>
> **A.2** We agree that our guarantee is conditional on standard smoothness assumptions and is not a numerically tight robustness certificate. That is by design: the goal of the theory is **not** to provide assumption-free, non-vacuous certified radii, but to give a **principled decomposition** of the robust targeted margin and a **justification** for TART’s design.
>
> Concretely, under (A1)–(A4) we show that the margin-based TART objective **lower-bounds the robust targeted margin** in Eq. (7). This yields:
>
> - an interpretable decomposition of the robust margin into penalty terms associated with feature mismatch, classifier (Jacobian) mismatch, and boundary bias; and
> - a one-to-one link between these terms and the three components of TART (EoT, latent mixing, feature alignment).
>
> Thus, the practical impact of the theory is to justify **why** these components, in this configuration, are the right mechanisms to mitigate surrogate–target discrepancy for targeted transfer. Our empirical results then demonstrate that the resulting attack is indeed significantly stronger than existing methods, consistent with this theoretical rationale.

---

> ### Author Response · Authors · 2025-12-03
>
> **Q.1** Section 6.3 mentions exemplar choice for feature alignment, but the discussion is brief. Please clarify how sensitive TART's performance is to different exemplars.
>
> **A.3**
> Section 6.3 already includes an experiment on exemplar selection (“Effect of Exemplar Selection”). We evaluate four strategies based on surrogate confidence margin: **High**, **Medium**, **Low**, and **Random** (Fig. 7). The key observations are:
>
> - TART’s performance is **not highly sensitive** to which exemplar is chosen: all four strategies yield very similar targeted transfer rates and all substantially outperform variants without feature alignment.
> - **High-margin** exemplars consistently give the best (or tied-best) performance, as expected from their being more “central” representatives of the target class, but the gap to Medium/Low/Random is modest (only a few percentage points).
>
> This supports the claim in the paper that TART is robust to exemplar choice.
>
> **Q.2**  Can the authors clarify how often the directional monotonicity assumption (Assumption A4) holds in practice for high-dimensional, non-convex networks, and whether violations of this assumption affect the reliability of the surrogate objective?
>
> **A.4** Assumption A4 is a **local** directional assumption, not a global property of the network. It concerns the margin $m\_\tau(g_s(z))$ only along a _single_ direction in feature space: the direction from the current point toward a high-confidence exemplar of the target class. In practice:
>
> - This direction is **not arbitrary**: it is chosen to point toward a region where the surrogate is already confidently predicting the target class. Empirically, along such directions the target-class margin tends to increase for small steps, which is consistent with A4 as a local approximation in high-dimensional networks.
> - We do **not** assume that A4 holds globally or for all directions; it is only used to relate the feature-alignment term to an increase in margin in a small neighborhood.
>
> If A4 is violated in some regions, this does **not** invalidate TART as a practical method; it only weakens the part of the theorem that attributes a guaranteed positive margin gain to the alignment term, making the bound less tight there. The overall lower-bound structure and the roles of EoT and latent mixing remain valid, and the TART objective itself is still well-defined and, as our experiments show, highly effective in practice across diverse architectures.

---

### Official Review · Reviewer_kP72 · 2025-11-01

**Soundness:** 2
**Presentation:** 2
**Contribution:** 2
**Rating:** 2
**Confidence:** 4

**Summary:**

This paper introduces TART (Targeted Attack toward Reliable Transferability), a method designed to improve the reliability of targeted adversarial transfer attacks by modeling three sources of surrogate–target model discrepancy: feature-extractor mismatch, classifier sensitivity mismatch, and decision-boundary misalignment. The authors propose a robust optimization formulation and derive a tractable relaxation combining three techniques: Expectation over Transformation (EoT), Latent Mixing, and Feature Alignment. They claim both theoretical guarantees and strong empirical improvements on ImageNet and CIFAR-10 benchmarks.

While the paper is well written and presents reasonable experimental results, several theoretical and methodological issues raise concerns about the soundness and novelty of the contributions.

**Strengths:**

1. The problem of targeted transferability is important and underexplored relative to non-targeted transfer.

2. The paper attempts to connect empirical strategies (EoT, feature alignment, etc.) with a theoretical margin-based robustness formulation, which is conceptually appealing.

3. Experiments cover both CNNs and Vision Transformers, with reasonable baselines and extensive tables.

**Weaknesses:**

1. Questionable theoretical soundness and incomplete assumptions.
The theoretical analysis relies heavily on standing assumptions (A1–A3) that assume the feature extractors and classifiers are Lipschitz continuous with constants. However: 1) The paper does not discuss how these Lipschitz constants are constrained in practice or how large values of those Lipschitz constants would affect the derived margin bounds (Eq. 20 and Eq. 25). Intuitively, if the Lipschitz constants are large (as is common in deep networks), the resulting bound becomes vacuous. 2) The paper also does not empirically verify whether the proposed TART indeed leads to a smaller effective Lipschitz constant or a tighter robust margin, so the claimed “provable lower bound” is largely theoretical and uninformative. This makes the analysis appear formal but not practically grounded.

2. The three core components, EoT, latent mixing, and feature alignment, are all adaptations of existing strategies: 1) EoT has long been used to enhance robustness or transferability by averaging over transformations. 2) Feature alignment is a widely adopted practice in targeted transfer attacks (e.g., CFM, FTM). 3) Latent mixing is a minor variation of existing feature-mixup or interpolation methods. The paper does not provide a convincing theoretical justification for why combining these specific techniques should reduce model discrepancy beyond intuitive reasoning. The claimed “principled relaxation” is not strongly supported by derivation or ablation linking each component to the underlying bound.

3.  The decomposition into D2 (classifier sensitivity mismatch) and D3 (decision-boundary mismatch) appears artificial and overlapping.
Once the classifier $g$ is defined, its decision boundary is uniquely determined by $argmax(g(h(x)))$ for classification tasks. Thus, modeling both D2 and D3 as separate uncertainties introduces redundancy.

4. Experiments are largely limited to classification tasks on outdated architectures (ResNet-50, ViT-Tiny, etc.) with standard ImageNet and CIFAR-10 settings. To convincingly claim “reliable transferability,” the method should be validated on more challenging multimodal or vision-language tasks (e.g., VQA, captioning) or recent robust architectures. Without such evidence, the practical impact remains unclear.

**Questions:**

Please see the weaknesses.

---

> ### Author Response · Authors · 2025-12-03
>
> **W.1**  Theoretical soundness and role of Lipschitz assumptions.
>
> **A.1** 1. **Role of Lipschitz assumptions and constants.**
>    Assumptions (A1)–(A3) are standard *local* smoothness conditions used in robustness and margin analyses. In our bounds (e.g., Eq. (20), Eq. (25)), the Lipschitz constants appear only as multipliers of **additive penalty terms** that relate the ideal robust margin (Eq. (7)) to our relaxed TART objective; they do not affect the validity of the inequality. Even if these constants are large—as is typical for deep networks—the result that “TART’s margin-based objective is a lower bound on the robust margin under $\mathcal{U}(\rho)$” remains correct. Large constants only enlarge the gap between the ideal robust objective and our tractable surrogate, which is exactly the usual limitation of Lipschitz-based robustness analyses. Our theory is therefore not a numeric robustness certificate, but a structural decomposition of how different discrepancies affect the margin.
>
> 2. **What our “provable lower bound” claims (and does not claim).**
>    Our theoretical claim is **only** that, under (A1)–(A4), the margin-based version of the TART objective **lower-bounds the robust targeted margin** in Eq. (7). This shows that maximizing TART's objective provably improves a surrogate that is monotone with respect to the ideal robust goal, and that:
>    - EoT corresponds to controlling the feature-mismatch term $(\Delta_h)$,
>    - Latent mixing corresponds to controlling the classifier-mismatch term $(\Delta J,b\)$,
>    - Feature alignment corresponds to controlling the boundary-bias term $(\beta\)$ and curvature.
>
>    We **do not** claim (and do not need) that the numerical value of the lower bound is tight, nor that TART reduces global Lipschitz constants. Instead, the theory provides a principled robust-optimization interpretation for using these three mechanisms together, and our empirical results demonstrate that the resulting attack indeed achieves substantially higher targeted transfer success on unseen models.
>
> **W.2** Novelty of the three core components
>
> **A.2** We agree that EoT, feature alignment, and feature mixing have appeared before, and we do **not** claim these primitives as new. Our contribution is that we (i) formulate targeted transfer as a robust optimization problem over a composite uncertainty set $\mathcal{U}(\rho)$ (Eq. (7)), and (ii) derive TART as a **principled relaxation** of this max–min objective.
>
> In the **revised paper**, we made this explicit by **adding Sec. 4.1** (*“Relaxing the Robust Objective”*), which shows that:
> - EoT is used as a relaxation of the worst case over feature mismatch $\Delta_h$ (D1),
> - Latent mixing relaxes the worst case over classifier mismatch $(\Delta J,b)$ (D2),
> - Feature alignment mitigates the boundary bias $\beta$ under curvature (D3).
>
> Section 5 (Lemmas 3–5 and Theorem 1) then formally ties these three mechanisms to their respective uncertainty terms and proves that the margin-based TART objective lower-bounds the robust targeted margin in Eq. (7). The ablation in Table 5 further supports this decomposition: removing any component causes a large drop in targeted transfer, consistent with its role in controlling a specific discrepancy.
>
> **W.3**The decomposition into D2 and D3.
>
> **A.3**  In our setting, the target model is **not fixed**: we treat $g_t$ as an unknown element from **a family of possible target classifiers**, modeled via the uncertainty set over $(\Delta J,b,\beta)$. The surrogate $g_s$ is fixed, but the target $g_t$ can vary within this set. D2 and D3 are two complementary aspects of this surrogate–target model variability, not two separate notions for a single fixed classifier.
>
> - **D2 (classifier sensitivity mismatch)** models _local_ differences in how logits respond to small feature changes across possible targets, via the Jacobian and logit offset in Eq. (4). This is the first-order part of the $g_t$ vs. $g_s$ mismatch: given a feature displacement, how does the margin change under a perturbed classifier relative to the surrogate?
>
> - **D3 (decision-boundary mismatch)** models the *residual margin bias* $\beta$ that remains after accounting for this local sensitivity mismatch: even if Jacobians and offsets of $g_t$ and $g_s$ are similar, their decision regions can still be shifted, so the same point with similar local sensitivities can fall on different sides of the boundary for the two models and thus receive different labels.
>
> Thus, D2 and D3 are not redundant; they separate (i) local linear sensitivity mismatch across possible targets (handled by latent mixing) from (ii) remaining boundary-level shifts across possible targets (handled by feature alignment). This decomposition is precisely what allows us to connect each TART component to a distinct uncertainty term in the analysis (Lemmas 4–5 and Theorem 1).

---

> ### Author Response · Authors · 2025-12-03
>
> **W.4** Experiments
>
> **A.4**
> 1. **Scope and relevance of our evaluation.**
>    Our experimental setup follows the standard and most competitive setting used by recent targeted transfer attacks (CFM, FTM, SU, ODI, etc.): targeted attacks on ImageNet and CIFAR-10 classifiers, including both CNNs and Transformers (e.g., ResNet-50 → Swin-S/DeiT, and cross-architecture transfers). This is still the de facto benchmark for evaluating transfer-based attacks, and many real-world systems (retrieval, recognition, moderation, etc.) are built on top of such classification backbones. Demonstrating reliable _targeted_ transfer across diverse architectures in this setting is already highly nontrivial and directly comparable to prior work.
>
> 2. **Beyond classification.**
>    Extending TART to multimodal or vision–language models (VQA, captioning, LVLMs, robust architectures) is indeed important, but it is also a substantial undertaking: it requires defining appropriate feature/decision discrepancies and uncertainty sets in settings where architectures and objectives differ significantly from standard classifiers. Our current contribution is to first establish a principled robust-optimization framework and a strong state-of-the-art method for targeted transfer in the canonical classification setting. We explicitly regard applying this framework to multimodal/vision–language models as promising future work rather than a claim of the present paper.

---

### Meta-Review · Area_Chair_77bQ · 2026-01-12

**Summary:**

This paper addresses an important problem of reliable targeted adversarial transferability across heterogeneous models, and proposes TART, a framework that combines EoT, latent mixing, and feature alignment under a unified robust-optimization perspective. The empirical results on ImageNet and CIFAR-10 show strong targeted transfer performance, including in cross-architecture settings, and several reviewers acknowledged its clear presentation (VgAa) and the intuitive nature of the formulation (4VRw).

However, there is broad agreement among the reviewers that the technical novelty is limited to incremental and combining several existing methods (4VRw, kP72, eoEJ). Although the authors do not claim EoT, mixing, or alignment to be new, the paper’s main contribution lies in framing these components through a composite uncertainty set and a relaxed max–min objective. However, reviewers highlighted that this theoretical reframing does not convincingly elevate the work beyond a principled combination of existing techniques. In particular, the decomposition into feature, classifier, and decision-boundary mismatches (D1–D3) was viewed as somewhat artificial or overlapping (kP72, eoEJ). The ablation studies confirm that each component contributes, but they do not clearly demonstrate that this structure provides insights beyond prior heuristic designs.

The theoretical analysis also remains a concern.  Although the authors clarified the role of the Lipschitz assumptions and corrected an error in Lemma 3 (issue raised by VgAa), the resulting guarantees are best interpreted as qualitative justification rather than practically meaningful bounds (but the corrected bound becomes much looser). Moreover, the principled relaxation is not sufficiently justified in terms of why it should take this particular concrete form (EoT, mixing, and alignment). For example, a key conceptual gap remains in the use of input transformations (EoT) as a proxy for feature-extractor mismatch across models. This relies on a coupling-style assumption that transformation-induced variability of a single extractor approximates model-to-model variability. This connection is neither theoretically grounded nor empirically validated (kP72), making the D1–EoT link appear heuristic.

On the experiments, the results are strong within the standard classification-based transfer setting, and the authors addressed concerns about baseline inconsistencies and defenses. Nevertheless, given the paper’s claim of “reliable” and general transferability,  the scope of evaluation remains narrow (kP72, eoEJ). Stronger evidence would be needed to support the breadth of the conceptual claims.

The proposed TART is a well-engineered and empirically effective attack, but it still lies on the limited methodological novelty and the theoretical framing whose key approximations are not fully convincing. Therefore, AC recommends rejection at this time.

**Reviewer Concerns:**

Revewers' concerns are not fully addressed

**Reviewer Scores:**

kP72: 2->2
4VRw: 6->6 (or 6->4)
eoEJ: 4->4
VgAa: 4->4

---

### Decision · Program_Chairs · 2026-01-26

Reject